# Small High-Risk Uveal Melanomas Have a Lower Mortality Rate

**DOI:** 10.3390/cancers13092267

**Published:** 2021-05-08

**Authors:** Rumana N. Hussain, Sarah E. Coupland, Helen Kalirai, Azzam F. G. Taktak, Antonio Eleuteri, Bertil E. Damato, Carl Groenewald, Heinrich Heimann

**Affiliations:** 1Liverpool Ocular Oncology Centre, Liverpool University Hospitals NHS Foundation Trust, Liverpool L7 8XP, UK; carl.groenewald@liverpoolft.nhs.uk (C.G.); heinrich.heimann@liverpoolft.nhs.uk (H.H.); 2Liverpool Ocular Oncology Research Centre, Department of Molecular and Clinical Cancer Medicine, University of Liverpool, Liverpool L69 7ZX, UK; s.e.coupland@liverpool.ac.uk (S.E.C.); h.kalirai@liverpool.ac.uk (H.K.); afgt@liverpool.ac.uk (A.F.G.T.); eleuteri@liverpool.ac.uk (A.E.); 3Department of Medical Physics and Clinical Engineering, Royal Liverpool University Hospital, Liverpool L69 8ZX, UK; 4Ocular Oncology Service, Moorfields Eye Hospital, London EC1V 2PD, UK; bertil.damato@ndcn.ox.ac.uk

**Keywords:** uveal melanoma, monosomy 3, metastatic risk

## Abstract

**Simple Summary:**

The current paradigm concerning metastatic spread in uveal melanoma is that the critical point for dissemination occurs prior to presentation and that treatment of the primary tumor does not change outcome. However, we show that patients with small uveal melanomas with genetic characteristics typical for high risk for metastatic disease have a lower mortality rate from metastatic disease, if treated earlier. Our data demonstrate that such small melanomas are potentially lethal (like larger tumors), but that there is a window of opportunity to prevent life-threatening metastatic spread if actively treated, rather than being monitored, as is often done currently.

**Abstract:**

Our aim was to determine whether size impacts on the difference in metastatic mortality of genetically high-risk (monosomy 3) uveal melanomas (UM). We undertook a retrospective analysis of data from a patient cohort with genetically characterized UM. All patients treated for UM in the Liverpool Ocular Oncology Centre between 2007 and 2014, who had a prognostic genetic tumor analysis. Patients were subdivided into those with small (≤2.5 mm thickness) and large (>2.5 mm thickness) tumors. Survival analyses were performed using Gray rank statistics to calculate absolute probabilities of dying as a result of metastatic UM. The 5-year absolute risk of metastatic mortality of those with small monosomy 3 UM was significantly lower (23%) compared to the larger tumor group (50%) (*p* = 0.003). Small disomy 3 UM also had a lower absolute risk of metastatic mortality (0.8%) than large disomy 3 UM (6.4%) (*p* = 0.007). Hazard rates showed similar differences even with lead time bias correction estimates. We therefore conclude that earlier treatment of all small UM, particularly monosomy 3 UM, reduces the risk of metastatic disease and death. Our results would support molecular studies of even small UM, rather than ‘watch-and-wait strategies’.

## 1. Introduction

The success of local control of primary uveal melanoma (UM) with radiotherapy or surgery is high (80–100%, depending on treatment modality) [1,2,3,4,5,6]. Recent advances in treatment regimens and the advent of intraocular laser therapies and drugs have increased the rates of eye- and vision retention [7,8,9,10,11]. Although certain treatments for metastatic UM, such as monoclonal antibody infusions or hepatic resections of isolated lesions, have shown some benefit [12,13,14,15,16,17,18,19,20,21], they are suitable for only select UM patients [22,23,24,25]. Clinical trials are underway for other treatments for patients with metastatic disease [26]; however, to date, the prognosis remains poor and the metastatic mortality rate associated with UM remains at 40–45% [27,28,29,30,31].

Prognostic parameters associated with an increased risk of developing metastatic UM are related to primary tumor characteristics, such as larger size, ciliary body involvement, extra-ocular extension and histologic features (e.g., epithelioid cell morphology, closed connective tissue loops, high mitotic count) [32,33,34]. In addition, chromosomal alterations, particularly loss of one copy of chromosome 3 (monosomy 3; M3) and gene mutations (e.g., in *BAP1, SF3B1* and *EIF1AX*) play a major role in determining metastatic risk in UM [35,36,37,38,39,40,41] [33]. At the Liverpool Ocular Oncology Centre (LOOC), metastatic risk is determined using the Liverpool Uveal Melanoma Prognosticator Online (LUMPO) algorithm that incorporates clinical, histological and genetic parameters, to provide an individualized approach to patient management [33]. This prognostic algorithm has been validated externally in multiple centers worldwide [42,43,44].

Many UM patients present with advanced ocular disease, with approximately 23% reporting that their tumor was initially missed when they presented with symptoms [45,46]. Previously, it had been considered that systemic tumor cell dissemination occurs prior to presentation, and hence that ocular treatment does not reduce the risk of metastatic disease and mortality. However, there are indications that earlier detection and treatment of smaller UM are not only associated with better local outcomes but may also be associated with improved survival rates [47,48,49,50,51,52,53]. 

Herein, to address the question of whether earlier treatment of UM would result in a reduction in mortality from metastatic melanoma, we present an analysis of the outcomes of UM patients with high-risk monosomy 3 UM (M3-UM), comparing their absolute risks of mortality according to tumor size. Our hypothesis is that earlier detection and treatment of high-risk M3-UM may improve the patients’ chances of survival. 

## 2. Results

During the 7-year study period, data were available for 940 patients whose UM underwent genetic testing. Of these, 403 (43%) were classified as M3; 432 (46%) as D3; and 105 (11%) were unclassifiable. The median age at diagnosis was 61 years (mean 60; range 24–94 years) with a male: female ratio of 525:415 (1.26:1). The median tumor diameter was 13.3mm (mean 14.3; range 1.8–26.0) and tumor thickness was 5.4mm (mean 6.5; range 0.7–18.3). Ciliary body involvement was present in 188 patients (Table 1).

Tumors were classified as ‘small’ (≤2.5 mm in thickness) (*n* = 196) and ‘large’ (>2.5 mm in thickness) (*n* = 744). This size cut-off is based on the clinical risk factors previously described for small melanocytic lesions [54] of 2 mm and the T1 staging of small melanomas of 3 mm [55]. 

As outlined in Table 1, primary treatment was enucleation in 407/940 (43%) patients; proton beam radiotherapy in 197 (21%), ruthenium-106 plaque brachytherapy in 222 (24%); photodynamic therapy in 2 (0.2%); trans-scleral local resection in 68 (7%); and transretinal endoresection in 42 (4%) patients. One case in each group had incomplete data on primary treatment. There were proportionally more cases treated with eye-conserving primary radiotherapy (either ruthenium-106 plaque brachytherapy or proton beam radiotherapy) in those with smaller tumors (72% vs. 37%, *p* < 0.00001), and higher rates of primary enucleation in the larger UM group (50% vs. 18%, *p* < 0.00001).

Of the small category tumors, 122 UM were D3 (62%), 44 were M3 (22%), and 30 (15%) were genetically unclassifiable. Of the large tumor group, 310 were D3-UM (42%), 359 were M3 (48%) and 75 (10%) were genetically unclassifiable. Both the small UM D3 and M3 subgroups were compared to the larger UM D3 and M3 cohorts, in terms of absolute risk of mortality due to metastasis (Figure 1). The Gray statistics showed significant evidence of an absolute risk difference between small and large M3-UM (*p* = 0.003) and between small and large D3-UM (*p* = 0.007). The small and large UM categories were also compared in terms of absolute risk of mortality (Figure 2). The Gray statistics (stratified by genetic class) also showed significant evidence of an absolute risk difference between small and large UM (*p* < 0.001). 

In particular, the 5-year absolute risk of mortality of small M3-UM was 23% versus larger M3-UM at 50% (small D3-UM 0.8%; large D3-UM 6.4%). 

Log-rank analysis also demonstrated a significant difference in the hazard rates of small M3 versus large M3 UM (*p* = 0.003), between small D3 and large D3 UM (*p* = 0.007) and between each of the remaining subgroup comparisons (*p* < 0.001). 

The relative risk (RR) of death over 5 years was 0.45 (95% CI 0.26–0.8) in the smaller M3 UM group compared to the larger M3 group, and the hazard ratio (HR) was 0.44 (95% CI 0.30–0.58). Assuming a ‘sojourn time’ of 1 year for detectability to symptomatology to correct for lead time bias as previously described [52], the modified RR of death was 0.69 (95% CI 0.45–1.05) and the modified HR was 0.67 (95% CI 0.53–0.81) (Table 2). The rates did not change significantly after 5-year correction for length time bias at a range of time courses (Appendix A).

## 3. Discussion

Due to the peak of metastatic mortality occurring 2 years after enucleation for UM [56], early hypotheses (Zimmerman–McLean–Foster) suggested that enucleation itself may increase the risk of metastatic disease by systemic dissemination of tumor cells [57]. If this study had been performed in the 1970s, proponents of the Zimmerman–McLean–Foster hypothesis might have attributed the higher mortality in patients with larger tumors to the treatment of such tumors by enucleation. This theory has been largely refuted by the COMS study with similar mortality rates for those undergoing enucleation or plaque brachytherapy [58,59]. It has therefore been assumed that hematogeneous dissemination of UM cells occurs long before the primary tumor has been diagnosed [60]. Tumor doubling time calculations support this hypothesis by suggesting that the metastatic dissemination precedes the initial diagnosis and treatment [61,62]. Other studies suggest that most UM patients have circulating tumor cells (CTC), whereby the genotypic and phenotypic profiles of the CTCs match the primary tumor [62,63]. As such, there is a prevailing dogma that treatment of the primary UM is undertaken only to preserve vision (and, if possible, the eye) but will not prevent metastatic disease. Alternate opinions propose, however, that by “stemming the metastatic flow” of UM cells into the bloodstream, this could result in a reduction in the subsequent CTC load, and consequently reduce metastatic risk [50,51,64]. That is, there may indeed be a window of opportunity for metastatic disease prevention by treating UM earlier, and this would be particularly relevant in small lethal M3-UM [53].

It is well established in UM that various clinical risk factors, including tumor size, are related to the risk of metastatic disease and subsequent mortality, with small incremental increases in primary tumor size significantly affecting outcomes [47,65,66]. It has been assumed that larger UM are innately more aggressive, as they have either high-risk genetic factors from the outset or have accumulated these factors over time, termed ‘crescendo malignancy’ [67]. On this basis, it is conventional practice in many ocular oncology centers to monitor small, indeterminate uveal melanocytic tumors for months (or even years) until growth is documented [68,69,70,71,72]. Weis et al. demonstrated that most of these indeterminate lesions are low-risk D3-UM [73]. Our current study supports these findings, with 73% of the small melanocytic tumors being D3; however, importantly, we demonstrate that over one quarter (i.e., 27%) of these small lesions are M3-UM, and therefore have a high metastatic risk. Furthermore, whilst most of these small UM are D3 at this stage, it is unclear what proportion could subsequently transform into higher-risk M3-UM, if left untreated. Certainly, there are reports of genetic heterogeneity within larger UM [67,74,75,76,77], which suggests that there is an evolutionary process of uveal melanocytic lesions from low to high genetic risk. In addition, we have previously demonstrated that asymptomatic patients with UM identified via the annual UK national diabetic retinopathy screening program have lower mortality than those detected via alternative routes [52]. Taking all of these data into account, which suggest that early detection and treatment of UM may enable the prolongation of life in patients, there could be justification for the consideration of earlier treatment in small uveal melanocytic tumors. If these lesions are monitored, adequate patient counselling and informed consent are vital. Afshar et al. described a number of cases whereby patients were monitored for years by their optometrist or local ophthalmologist without being informed that there was a possibility that their ‘suspicious nevus’ was malignant; some of these patients were eventually found to have a lethal UM when biopsy was ultimately performed [78]. On the other hand, it is also important to emphasize here that we do not recommend treatment of all small, benign, melanocytic lesions; it can indeed be difficult to differentiate benign nevi from small melanomas. These are best assessed and treated by teams having the required expertise in ocular oncology so as to avoid unnecessary visual loss whilst not delaying treatment of life-threatening melanomas. For example, we were able to confirm malignancy in such small lesions by performing biopsy using skills and techniques that are now widely available.

At present, unfortunately, it is not possible using the available clinical imaging technologies to genetically subtype these small uveal melanocytic lesions. With the advent of artificial intelligence and its application in ophthalmology, however, this may change [79]. Consequently, at present, to determine the genetic status of the cells within an atypical uveal melanocytic lesion, intraocular biopsies must be performed. Our previous experience indicates that this procedure is safe in most patients and can be undertaken following radiotherapy [80,81].

In this analysis, we have attempted to control for lead time bias with previously published methods [82] requiring an estimate of sojourn time, which is the unobserved period of time during which the tumor is asymptomatic but detectable at screening, and thus it artificially increases the follow-up time. Sojourn time estimation is a difficult problem in general, particularly in UM, where often asymptomatic patients are detected at a routine screening visit [45]. Therefore, we have created a range of estimates, as suggested for other forms of cancer where sojourn time is undefined [82], in order to take this into account, as previously described [52]. Although these calculations come with their limitations, these corrected calculations have also supported a risk benefit of those small melanoma lesions treated at an earlier stage.

A weakness of our study is that only chromosome 3 aberrations were assessed. We have demonstrated chromosome 3 status to be of the greatest predictive value in stratifying metastatic risk [83]; as such, we have used this parameter in these small samples. Other genetic alterations are well-known to contribute to an increased metastatic risk (e.g., gains of chromosome 8q, class 2 gene expression profiles and *BAP1* (BRCA associated protein 1) mutations) [84,85,86,87,88,89,90]. We and others have developed bespoke next-generation sequencing panels for UM requiring smaller DNA concentrations from small lesions, which can be applied to a range of processed and size-varying samples [91,92]. The application of this molecular panel in future UM cohorts may enable us to delve further into the genetic interactions and intricacies of each patient subgroup with different sized tumors. 

## 4. Methods

### 4.1. Study Design

We performed a retrospective case–control study of data from a cohort of UM patients diagnosed at the LOOC over the period of 2007–2014, to enable a minimum of 5 years of follow-up. All patients who had consented to genetic testing of their tumor were included. The diagnosis of UM was established by the presence of clinical signs of malignancy (e.g., orange pigment, low ultrasound reflectivity, serous retinal detachment and/or documented growth) and followed by treatment with local radiotherapy (with trans-scleral or trans-retinal biopsy), tumor resection or enucleation. We analyzed data on all our cases with a M3 or disomy 3 (D3) genetic result (see below). Tumors were classified as ‘small’ (≤2.5 mm in thickness) and ‘large’ (>2.5 mm in thickness). This size cut-off is based on the clinical risk factors previously described for small melanocytic lesions [54] of 2 mm and the T1 staging of small melanomas of 3 mm [55].

### 4.2. Clinical Methods

Date and certified cause of death were automatically provided by the National Health Service (NHS) Cancer Registry, the patients having been flagged at the time of diagnosis. International patients and those from Ireland were excluded from this study as their outcomes were not recorded within this NHS database.

### 4.3. Tumor Sampling

Tumor sample collection included material obtained during enucleation, trans-scleral tumor resection and primary tumor endoresection. Those undergoing radiotherapy as primary treatment (either proton beam or ruthenium-106 plaque brachytherapy) were biopsied by either the trans-scleral or trans-retinal route. Trans-retinal samples were obtained with a 25-gauge vitreous cutter, as previously described [93,94]. Trans-scleral biopsies were initially obtained with 25 g fine-needle aspiration (FNAB), then with Essen forceps scleral flap technique.

### 4.4. Genetic Analysis

Tumor DNA extraction, DNA quality assessment and quantification were conducted, as previously described [95,96]. Chromosome 3 aberrations were assessed by either multiplex ligation-dependent probe amplification (MLPA) (MRC-Holland, Amsterdam, The Netherlands) or microsatellite analysis (MSA). MLPA and MSA methods have been previously described [95,96,97,98,99].

### 4.5. Statistical Analysis

Gray rank statistics [100] were utilized to assess the differences in absolute risks of dying due to metastatic UM amongst the groups, enabling the consideration of competing risks. Hazard rates were also compared with the standard log-rank test. Lead time and length time biases were estimated according to the method described by Duffy et al. [82].

## 5. Conclusions

Our study has shown that treatment of small M3-UM is potentially lifesaving in its early stages. It is, therefore, necessary to identify and treat malignancy in these small, potentially lethal lesions without delay, if necessary, after performing intraocular tumor biopsies. There appears to be a survival benefit in treating all small UM to reduce the flow of metastatic cells and, more controversially, perhaps also small D3 UM to prevent conversion from low- to high-risk tumor characteristics, such as in those cases of developing genetic heterogeneity. This challenges the prevailing paradigm that clinically significant micrometastases occur prior to diagnosis of the ocular tumor, and suggests that a window of opportunity exists to reduce the flow of UM micrometastases and subsequent incurable, lethal dissemination. This window may only be available in the early stages of tumor development. Consequently, these small, lethal M3-UM may represent the only melanocytic lesions for which ophthalmologists can indeed alter patient mortality outcome with early treatment.

## Figures and Tables

**Figure 1 cancers-13-02267-f001:**
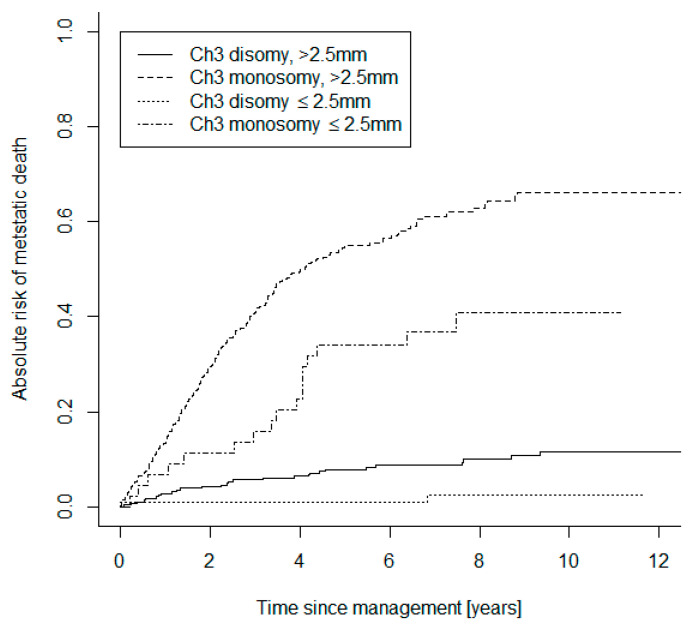
Curves showing the comparative absolute risks of mortality due to metastatic UM in monosomy 3 and disomy 3 tumors subdivided into the defined ‘small’ and ‘large’ categories.

**Figure 2 cancers-13-02267-f002:**
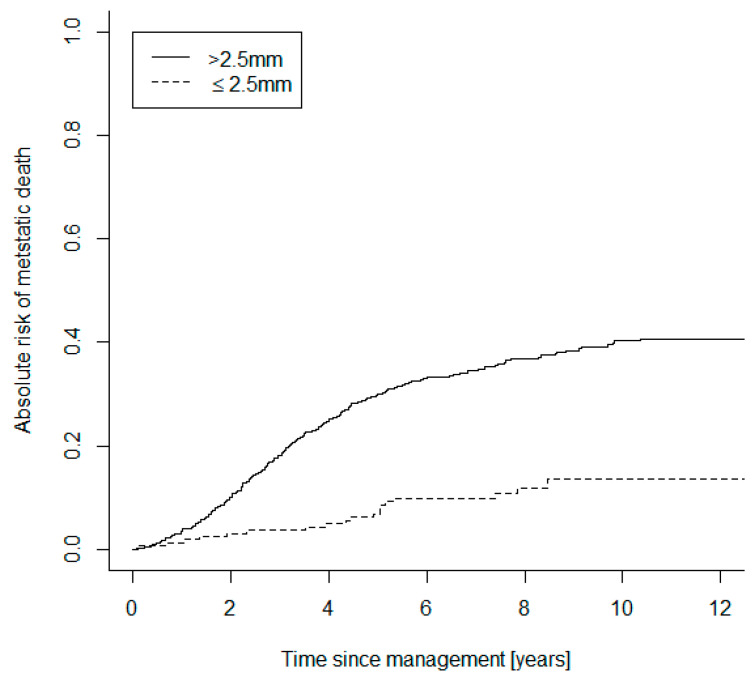
Curves showing the comparative absolute risks of mortality due to metastatic UM subdivided into the defined ‘small’ and ‘large’ categories.

**Table 1 cancers-13-02267-t001:** Baseline patient and tumor characteristics (median and range) and primary treatment of small and large tumor categories.

Baseline Characteristics and Treatments		Small UM Group (*n* = 196)	Large UM Group (*n* = 744)
Baseline characteristics	Age at diagnosis (years)	59 (22–89)	62 (21–94)
	Longitudinal base diameter (mm)	8.5 (1.8–19.8)	14.2 (2.3–26)
	Tumor thickness (mm)	1.8 (0.7–2.5)	6.8 (2.6–18.3)
	Ciliary body involvement	17 (8.7%)	169 (22.7%)
Primary treatment	Ruthenium-106 plaque brachytherapy	77 (39.2%)	145 (19.5%)
	Proton beam radiotherapy	65 (33.2%)	132 (17.7%)
	Trans scleral tumor resection	8 (4.1%)	60 (8.1%)
	Endoresection	9 (4.6%)	33 (4.4%)
	Primary enucleation	35 (17.9%)	372 (50.0%)
-	PDT	1 (0.5%)	1 (0.1%)

**Table 2 cancers-13-02267-t002:** Risk ratio and hazard rates of death for small M3 tumors in comparison to larger M3 UM. Lead time bias correction with an estimated sojourn time of 1 year still demonstrates a lower risk of death due to metastatic disease.

Risk Ratio and Hazard Rates	RR (95% CI)	HR (95% CI)
Uncorrected	0.45 (0.26–0.80)	0.44 (0.30–0.58)
After lead time bias correction	0.69 (0.45–1.05)	0.67 (0.53–0.81)
After length time bias correction after 5 years	0.78 (0.69–1.01)	0.64 (0.5–1.03)

## Data Availability

Data are available on request on a secured database within the Oncology Unit.

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
