# Peer review of "Small High-Risk Uveal Melanomas Have a Lower Mortality Rate"

_cancers, 2021, doi:10.3390/cancers13092267_

Round 1

Reviewer 1 Report

Dear Authors,

your manuscript has improved considerably.

In my opinion  your paper is now acceptable for publication. 

Congratulations!

Reviewer 2 Report

The authors have appropriately addressed the comments raised by the reviewers. 

From my point of view, the second sentence of the discussion may be rephrased as it is irritating in its present form. 

Reviewer 3 Report

Having read the sent file I consider this manuscript ready for publication.

This manuscript is a resubmission of an earlier submission. The following is a list of the peer review reports and author responses from that submission.

Round 1

Reviewer 1 Report

In this study authors determine whether size impacts on the difference in metastatic mortality of genetically high risk (monosomy 3) uveal melanomas. It is an interesting study but as pointed out by authors has a serious weakness that only chromosome 3 aberrations were assessed. Other genetic alterations  (e.g. gains of chromosome 8q, class 2 gene expression profiles), inactivating mutations of BAP1 and histological features (epithelioid type, mitoses, extravascular matrix pattern, tumour infiltrating macrophages, etc), and clinical characteristics (ciliary body involvement, extraocular extension, etc) are well-known to contribute to an increased metastatic risk should be analysed. In my opinion, additionally, a multivariate analysis seems to be in order.

Reviewer 2 Report

The authors analyse a large number of uveal melanomas with genetic information and adequate follow-up for differences in tumor size.

As the authors state, this is an important work as it shows that small M3 UMs have a better outcome than large M3 UMs supporting early treatment. The study is well conducted, well written and comes from well-respected experts in UM. It may have a sustainble impact on the treatment on UM and will stimulate the discussion on early treatment on UM.

I have only minor comments:

  • It should be highlighted that it is important to clearly distinguish between nevi and small UM since the treatment of small UM (in most cases brachytherapy or protom beam irradiation) has often vision-affecting side effects. It may be difficult in some cases to distinguish between nevi and small UM clinically and I am sure that the authors agree that the study should not result in an overtreatment of nevi.
  • Since large M3 UMs were treated more often by enucleation than small UMs, I recommend to briefly discuss the Zimmerman-McLean-Foster hypothesis on the suggested relation between enucleation and metastatic disease.
  • The sentence (line 121-125) may need rephrasing.

Reviewer 3 Report

I read the paper entitled “High risk uveal melanomas have a lower mortality rate if treated early” analysing correlation between thickness  of  monosomy 3 uveal melanoma and metastatic mortality with interest, and I would like to the following questions to its authors:

  1. What was an absolute risk of mortality due to metastasis between small and large UM independent of chromosome 3 status?
  2. Did the authors perform a statistic analysis based not only on thickness of the lesion but also largest basal diameter?

Moreover references to the literature need to be reorganized.